# Activation of mTOR Signaling Pathway in Hepatocellular Carcinoma

**DOI:** 10.3390/ijms21041266

**Published:** 2020-02-13

**Authors:** Gustavo Ferrín, Marta Guerrero, Víctor Amado, Manuel Rodríguez-Perálvarez, Manuel De la Mata

**Affiliations:** 1Instituto Maimónides de Investigación Biomédica de Córdoba (IMIBIC), Universidad de Córdoba, 14004 Córdoba, Spain; gusfesa@gmail.com (G.F.); martagmisas1987@hotmail.com (M.G.); victoramadotorres@hotmail.com (V.A.); mdelamatagarcia@gmail.com (M.D.l.M.); 2Centro de Investigación Biomédica en Red de Enfermedades Hepáticas y Digestivas (CIBERehd), 14004 Córdoba, Spain; 3Department of Hepatology and Liver Transplantaton, Hospital Universitario Reina Sofía, 14004 Córdoba, Spain

**Keywords:** mTOR, everolimus, sirolimus, hepatocellular carcinoma, liver transplantation, sorafenib

## Abstract

Hepatocellular carcinoma (HCC) is the most frequent primary liver cancer and occurs mainly in patients with liver cirrhosis. The mammalian target of rapamycin (mTOR) signaling pathway is involved in many hallmarks of cancer including cell growth, metabolism re-programming, proliferation and inhibition of apoptosis. The mTOR pathway is upregulated in HCC tissue samples as compared with the surrounding liver cirrhotic tissue. In addition, the activation of mTOR is more intense in the tumor edge, thus reinforcing its role in HCC proliferation and spreading. The inhibition of the mTOR pathway by currently available pharmacological compounds (i.e., sirolimus or everolimus) is able to hamper tumor progression both in vitro and in animal models. The use of mTOR inhibitors alone or in combination with other therapies is a very attractive approach, which has been extensively investigated in humans. However, results are contradictory and there is no solid evidence suggesting a true benefit in clinical practice. As a result, neither sirolimus nor everolimus are currently approved to treat HCC or to prevent tumor recurrence after curative surgery. In the present comprehensive review, we analyzed the most recent scientific evidence while providing some insights to understand the gap between experimental and clinical studies.

## 1. Introduction

Hepatocellular carcinoma (HCC) is the most frequent primary liver malignancy and it accounts for 85–90% of all hepatic cancers. HCC is the fifth most prevalent cancer and the second cause of cancer-related death worldwide. The natural history of liver cirrhosis is tightly linked with HCC: a chronic liver insult (viral hepatitis, alcohol intake or non-alcoholic steatohepatitis (NASH) among others), which is able to produce inflammation of the liver parenchyma, which in turn triggers the activation of hepatic stellate cells and portal myofibroblasts, thus resulting in progressive liver fibrosis and, finally, in liver cirrhosis [1]. Both chronic inflammation and fibrosis are well known risk factors of cancer. NASH is also associated with an abnormal lipid metabolism, lipid accumulation within the hepatocytes, obesity and insulin resistance, with this abnormal metabolic scenario also able to promote cancer initiation and progression [2]. As a result, the vast majority of patients with HCC have an underlying chronic liver disease and indeed, HCC is the first cause of death among patients with liver cirrhosis.

Mutations in oncogenes and tumor suppressor genes are the main pathogenic mechanisms underlying hepatocarcinogenesis as they are responsible for disrupting cell signaling pathways. In HCC, the major pathways involved in the oncogenic process are WNT/b-catenin, Hedgehog, hepatocyte growth factor/c-MET, vascular endothelial growth factor (VEGF), mitogen-activated protein kinase (MAPK)/ERK (or Ras-Raf-MEK-ERK) and PI3K/AKT/mTOR [3,4]. Among all, the PI3K/AKT/mTOR pathway is particularly interesting because it is constitutively activated in a significant proportion of patients with HCC and it is associated with a more aggressive tumor progression and shorter survival [5].

In the present manuscript we aimed to review the role of the mammalian target of rapamycin (mTOR) pathway in the pathogenesis and biological aggressiveness of HCC to better understand the potential clinical impact of its inhibition by currently available drugs. In addition, we have provided some insights regarding the limitations to reproduce the beneficial effects of mTOR inhibitors observed in experimental models into human trials.

## 2. The mTOR Signaling Pathway

The mechanistic target of rapamycin (mTOR) is an evolutionary conserved serine/threonine kinase, which belongs to the phosphoinositide 3-kinase (PI3K)-related protein kinase family and integrates environmental information into cellular responses [6]. The main structure of the mTOR pathway and its role in liver diseases including HCC are summarized in Figure 1.

mTOR is the keystone of at least two structurally different multiprotein complexes, known as mTOR complex 1 (mTORC1) and mTORC2 [7,8,9]. Both mTOR complexes share the mammalian lethal with SEC13 protein 8, the Tti1/Tel2 complex and the inhibitory protein DEP domain-containing mTOR-interacting protein (DEPTOR). However, while the mTORC1 contains the regulatory-associated protein of mTOR (Raptor) and the inhibitory subunit proline-rich Akt substrate of 40kDa (PRAS40), the mTORC2 exhibit the rapamycin-insensitive companion of mTOR (Rictor) and the regulatory proteins Protor1/2 and mSin1. These structural particularities of mTORC1 and mTORC2 are responsible for a different response pattern to a wide variety of upstream signals, as well as for activating different downstream effectors, thus resulting in well differentiated cellular responses [10,11,12].

mTORC1 and mTORC2 respond to insulin and related growth factors through the interconnected MAPK/ERK and PI3K/AKT/mTOR signaling pathways [13,14]. Many growth factors are able to trigger these pathways by interacting with their cell-surface receptor tyrosine kinases (RTKs). The mTORC1, which is the best characterized complex, may be activated by nutrients, cytokines, oxygen and stress signals by different routes, among which adenine monophosphate (AMP) activated protein kinase (AMPK), which is an essential regulator of protein kinase B (or AKT), seems to be particularly relevant [15]. Under favorable conditions, mTORC1 activation promotes lipid and protein biosynthesis, cell growth and proliferation, while motivating the inhibition of autophagy [16]. On the other hand, mTORC2 exert its control on cell survival, metabolism and proliferation, as well as on the organization of the cell cytoskeleton, by controlling the activity of different protein kinases [17,18,19], including also AKT [20]. In addition, both mTORC1 and mTORC2 regulate each other by controlling the activity of ribosomal protein S6 kinase (S6K) and AKT respectively, although the extent of these mechanisms is not completely understood [21].

## 3. Regulation of the mTOR Pathway in Liver Diseases and HCC

The mTOR signaling pathway is frequently deregulated in cancer and metabolic diseases [22]. Some metabolic diseases such as diabetes or obesity are widely known risk factors of cancer in the general population. Indeed, as part of the oncogenic process, a deep transformation of the metabolic cell program is required, with completely different bioenergetic and biosynthetic demands [23]. Alcoholic liver disease (ALD), chronic viral hepatitis (B and C), NASH and, to a lesser extent, cholestatic liver diseases are the most common causes of liver cirrhosis worldwide and they are characterized by an abnormal regulation of the mTOR pathway, although with some peculiarities for each one of them.

The liver is a paramount organ for lipid metabolism [24]. During feeding, when carbohydrates are abundant, the liver inhibits gluconeogenesis and transforms glucose into fatty acids by de novo lipogenesis, which is regulated by insulin and requires the activation of the PI3K/AKT/mTOR-signaling pathway, together with the enzymes acetyl-CoA carboxylase and fatty acid synthase. Additionally, fatty acids from gastrointestinal digestion are also transported to the liver by the portal bloodstream. Under such conditions, mTORC1 inhibits catabolism (and activates anabolism) by suppressing autophagy through the inactivation of unc-51 like autophagy activating kinase (ULK1) [25] and transcription factor EB (TFEB) [26]. Therefore, fatty acids are processed and stored as triglycerides in lipid droplets whenever needed depending on the metabolic environment. Conversely, during fasting, the liver obtains energy in the form of glucose and fatty acids by hepatic gluconeogenesis and lipolysis, respectively. In addition, the induction of autophagy by the mTOR pathway has also been associated with lipolysis and with the mobilization of intracellular lipid stores from the liver [27].

Alcohol intake may disrupt normal hepatic lipid metabolism and may lead to fatty liver or steatosis by promoting hepatic lipogenesis while hampering fatty acids transport and oxidation. This metabolic reprogramming in alcoholic liver disease requires the activation of mTORC1 by downregulating DEP domain-containing mTOR-interacting protein (DEPTOR, an upstream inhibitor of the complex) [28], and by inhibiting AMPK, which results in the upregulation of lipogenic gene sterol regulatory element binding protein-1 (SREBP1), and in the reduction of circulating adiponectin and peroxisome proliferator activated receptor-alpha (PPAR-alpha) [29]. Chronic alcohol consumption is also associated with oxidative stress, inhibition of hepatocellular autophagy and apoptosis [30,31]. Moreover, oxidative stress induced by chronic alcohol consumption inhibits AMPK and activates the mTOR signaling that, in turn, inhibits autophagy [32]. In contrast, alcohol binging promotes autophagy as a protective mechanism of hepatocytes from direct alcohol toxicity and steatosis, probably through a reactive oxygen species (ROS)-dependent inhibition of mTOR activity [33]. Therefore, acute and chronic alcohol administration has a dual and paradoxical effect on the mTOR pathway to modulate hepatic autophagy. The differential nuclear content of TFEB could be partially responsible for the observed differences [34,35]. Recent studies have shown that miR-155 targets mTOR in chronic alcoholic liver disease to impair autophagy and lysosome function [36,37]. Alcohol-induced miR-155 downregulates key proteins of the autophagy process such as lysosome-associated membrane proteins LAMP1 and LAMP2 and, indirectly, TFEB. In turn, these proteins and lysosome dysfunction exerts feedback on the mTOR pathway to maintain and amplify liver damage [38].

Chronic infection by hepatitis B (HBV) and C (HCV) viruses is still a common cause of liver cirrhosis, although their prevalence is expected to continue to decrease in the upcoming years due to advances in antiviral therapy. The role of the mTOR pathway in chronic viral hepatitis concerns the regulation of cellular homeostasis. HBV and HCV use the mTOR pathway to enhance autophagy [39] to benefit their own replication [40]. A transgenic mice model with impaired autophagy is characterized by a strong reduction in HBV replication [41]. The activation of the PI3K/AKT/mTOR signaling pathway inhibits both autophagy and HBV transcription/replication in liver cancer cells and could be responsible for the elimination of viral replication that occurs during liver oncogenesis [42]. The link between hepatitis B/C, mTOR activation and autophagy is not fully understood but the miR-99 family members, which are aberrantly expressed in the serum of HBV patients, have been proposed as key regulators in this process [43].

NASH is the liver manifestation of metabolic syndrome and it is characterized by hepatic steatosis, ballooning, inflammation and progressive fibrosis. NASH occurs as a result of an imbalance of energy intake/consumption or when there is an intrinsic problem with the lipid storing process. Within the hepatocyte, the accumulation of fatty acids leads to the production of lipid-synthesis intermediates, which causes steatosis through the mTOR pathway deregulation [44]. Thus, NASH shares many mechanisms and pathological features with ALD. In a high-fed diet mice model of NASH, the pharmacological inhibition of mTORC1 or phosphatidylinositol 3-kinase (PI3K) protects against steatosis while promoting insulin sensitivity [31]. In a different murine model, Zhang et al. showed an initial upregulation of the mTORC1, with a subsequent damping oscillation pattern in the long term, being the alteration of the mTORC1 opposite to that observed for TFEB. These findings suggested that hepatic steatosis progresses by factors other than anabolic synthesis such as inhibition of autophagy and lysosome dysfunction [38]. Perinatal exposure to bisphenol has also been associated with the development of NASH in mice offspring through the activation of the PI3K/AKT/mTOR pathway in a process that involves the overexpression of lipogenic genes, autophagy impairment and inflammatory response [45]. On the other hand, dysbiosis of the gut microbiota may play a key role in the development of NAFLD [44]. Alterations in gut microbiota are able to disturb physiological functions controlled by mTOR such as lipogenesis, gluconeogenesis or autophagy. The use of mTOR inhibitors to influence gut microbiota in order to treat NASH is an attractive research field which should be further explored [46]. Obesity, which is a major contributor to NASH, enhances the development of chemically induced HCC through the activation of mTOR and the inhibition of AKT [47].

Cholestatic liver diseases result from a repeated exposure to unconjugated primary biliary acids, which may activate mTOR through the IκB kinase β (IKKβ)/Tuberous sclerosis 1 (TSC1)/mTOR pathway as demonstrated in some precancerous conditions such as Barrett’s esophagus [48]. In the liver, chronic cholestasis is a risk factor for HCC development and conversely advanced HCC usually provokes a progressive cholestatic effect. Glycochenodeoxycholic acid, which is a glycine-conjugated form of the primary bile acid chenodeoxycholic acid, is highly cytotoxic in a dose-dependent manner and its exposure in cellular models of the disease [49] activates cellular autophagy through the AMPK/mTOR pathway to promote HCC invasiveness [50]. Secondary biliary acids produced by the metabolic activity of the gut microbiota are also able to enhance AKT-independent mTOR activity in the liver to promote hepatic steatosis and HCC, while depleting gut microbiota with antibiotics reverses these mechanisms [51].

Liver cancer development in different mouse models with genetic or dietary alterations is frequently preceded by hepatic steatosis and is associated with a deregulation of the mTOR pathway [52,53,54,55,56]. A constitutive activation of the mTOR pathway may be enough to cause fatty liver and HCC independent of the diet. Moreover, mTOR pathway deregulation has been linked to liver cancer independent of the development of liver steatosis [57]. There is evidence of mTOR pathway activation in HCC, cholangiocarcinoma and hepatoblastoma [58]. This abnormal activation involves both PI3K/AKT and MAPK/ERK [59] signaling pathways, which are frequently co-induced in liver cancer. Thus, the mTORC1 and mTORC2 pathways are upregulated in 40–50% of patients with HCC [60,61]. In addition, increased mTOR expression is found in larger tumors or in advanced stages of HCC [62,63]. In a prospective study including 49 patients with HCC who underwent liver transplantation, the explanted liver was analyzed to determine the mTOR pathway expression both in the tumor and in the surrounding non-tumoral cirrhotic tissue [64]. mTOR pathway expression was more intense at the tumor edge where cellular growth and proliferation are more intense. Noteworthy, a subanalysis of patients who had undergone locoregional ablation of the tumor prior to liver transplantation revealed a significant reduction in mTOR activity within the tumor but not in the surrounding tissue. This suggests that local ablation therapies are not able to completely suppress mTOR pathway activity, thus explaining why tumor recurrence is almost universal if no other therapy is implemented afterwards. In light of this evidence, it is not surprising that a more intense mTOR pathway deregulation is associated with increased tumor recurrence rates after surgical resection or liver transplantation, as well as with shorter survival [64].

## 4. Targeting mTOR Signaling in HCC: Experimental Models

As described above, the mTOR pathway is systematically deregulated among the most prevalent etiologies of liver cirrhosis and it may be found overexpressed in patients with HCC, where it confers a more aggressive biological behavior. Therefore, a pharmacological and molecular inhibition of the mTOR signaling pathway has been proposed as a therapeutic strategy for HCC [11]. Next, we have described the pharmacological compounds with anti-mTOR activity currently available for patients with chronic liver disease, including HCC (Table 1). Other drugs or strategies which also target mTOR such as miRNAs [65], antibiotics [51] or antioxidants [66,67], but are not routinely used in clinical practice, are out of the scope of this review and have not been discussed here.

### 4.1. Ursodeoxycholic Acid (UDCA)

UDCA is a cytoprotective secondary bile acid which forms the backbone of the therapeutic arsenal against cholestatic liver diseases, mainly primary biliary cholangitis. UDCA therapy improves liver biochemistry parameters and delays the progression of the disease to liver cirrhosis [68]. In a human liver cancer cell line, UDCA induced the synthesis of glutathione by activating the PI3K/AKT pathway and displayed antioxidant properties [69]. In addition, UDCA could prevent lipid accumulation and reduce liver damage by inhibiting the AKT/mTOR/SREBP1 pathway [70]. Similarly, U12, an UDCA derivative compound, is able to hamper mTOR activity in vitro and in vivo and has shown antiproliferative and proapoptotic capacities, which could be helpful to prevent cancer [71]. Nor-UDCA, a more potent homologue with anti-inflammatory, anti-cholestatic and anti-fibrotic properties, induces autophagy by activating the AMPK/mTOR/ULK1 signaling pathway (i.e., suppressing mTOR activity) and has been suggested as a novel therapeutic approach to prevent fibrosis progression and HCC in patients with alpha-1 antitrypsin deficiency [72].

### 4.2. Rapalogs

Rapamycin (also known as sirolimus), an antibiotic first isolated in 1972 by Sehgal and Cols from the bacterium *Streptomyces hygroscopicus* [73], is a potent and selective inhibitor of the mTOR protein kinase. Rapamycin exerts its activity mainly on mTORC1, although a prolonged therapy could also disrupt mTORC2 to a lesser extent [74]. Rapamycin promotes the inhibition of TFEB [38], which is associated with autophagy and catabolic processes such as fatty acid oxidation and ketogenesis. Currently, rapamycin analogues (also known as rapalogs) are widely used to modulate autophagy in experimental models. Indeed, targeting mTOR is an attractive approach for liver diseases in which autophagy has protective effects, such as storage disorders (alpha-1 antitrypsin deficiency or Wilson’s disease), acute liver injury, alcoholic liver disease, NASH or HCC [31,33,75,76,77,78]. However, since autophagy has a dual role (beneficial or detrimental) depending on the cell type and the stage of the disease, it should be considered to target specifically liver cells while considering the optimal therapeutic window for its promotion (in earlier disease stages) or inhibition (in more advanced stages or in HCC) [79].

Rapamycin has shown potent antiproliferative and immunosuppressive properties against a large variety of tumor cells in vitro and suppresses growth of cancer cells in vivo [10,80,81,82,83]. Rapalogs with an improved pharmacokinetic profile and solubility properties have been tested in clinical trials. Temsirolimus and everolimus were approved for treatment of metastatic renal carcinoma among other malignancies. However, no rapalog has been approved for HCC treatment to date (see next section), probably due to the partial inhibition of mTORC1 and the collateral hyperactivation of the MAPK/ERK pathway through a PI3K-dependent feedback loop (mTORC1-MAPK feedback loop) [84]. In this sense, everolimus inhibits both mTOR complexes more potently than sirolimus, particularly mTORC2 [85], which could be an advantage in terms of drug efficacy and safety. In any case, most studies performed in animal models as outlined above implemented a local intra-tumor administration of the drug, which does not mirror clinical practice in humans. In those studies, with oral administration of mTOR inhibitors in murine models, the dosage was up to 100-fold increased as compared with the average dose in humans. This may explain in part the difficulties to reproduce the anti-proliferative properties of mTOR inhibitors in human clinical trials (see Section 5 below).

### 4.3. Second-Generation mTOR Inhibitors in HCC

mTOR kinase inhibitors (TOR-KIs) are second-generation mTOR inhibitors, which emerged to solve the above referred limitations of rapalogs. TOR-KIs appear to be more potent than rapalogs because they inhibit the activity and associated functions of both mTOR complexes (related to protein and lipid biosynthesis, cell growth and proliferation). Several of these compounds are currently being tested in preclinical experiments and early clinical trials [86,87,88]. However, there is a concern that TOR-KIs also cause feedback-dependent biphasic regulation of AKT signaling, which involves RTKs and causes the reactivation of mTOR signaling. Further investigations are warranted [89].

### 4.4. Sorafenib

Sorafenib is a multitarget kinase (multi-kinase) inhibitor that, among its multiple action mechanisms, interferes with the AKT/mTOR pathway by targeting VEGFR, PDFGR, c-Kit, c-RAF and B-RAF [90]. Sorafenib is currently the standard of care for patients with advanced HCC and for those patients with intermediate-stage HCC who are not eligible for locoregional therapies [91]. In liver cancer cells, sorafenib disrupts lipogenesis and provokes cell death by suppressing the production of ATP, which results in AMPK activation, mTOR inhibition and SREBP1 reduction [92]. It seems that AMPK blocks mTORC1 by phosphorylating the mTOR inhibitor TSC2 [93] and the mTORC1 subunit Raptor [94]. These findings agree with the established dependence of cancer cells on de novo fatty acids biosynthesis and enhanced lipogenesis. This pathway could represent a promising target for cancer therapy. Recently, regorafenib, another multi-kinase inhibitor, was approved as a second line therapy for HCC patients who experience tumor progression regardless of sorafenib. However, the clinical benefit of these multi-kinase inhibitors is modest because patients do not respond homogeneously, which could be partially explained by the development of drug resistance through the activation of the MAPK/ERK signaling pathway [95].

### 4.5. Combination of mTOR Inhibitors in HCC

The oncogenic signaling pathways of HCC are dynamic and able to adapt to new situations by creating compensatory mechanisms, which provide tumor cells with the ability to evade individual drugs. Accordingly, new approaches based on the combination of different inhibitors targeting complementary critical points of these interconnected pathways have been tested. Yu et al. [88] recently published a comprehensive review, which summarized clinical trials (completed or still ongoing) involving mTOR inhibitors alone or in combination in HCC. It not only included mTOR inhibitors such as rapalogs and TOR-KIs but also PI3K, AKT and PI3K/mTOR inhibitors.

Sorafenib produces heterogeneous cellular responses depending on the genomic context or mutational profile of the tumor. When sorafenib is used in combination with specific-targeted inhibitors driven from a close analysis of the mutational cell background, the observed effect was synergic regarding the inhibition of HCC proliferation. This fact could partially explain the suboptimal results of sorafenib, alone or in combination, in clinical trials where a heterogeneous and uncharacterized group of patients was included [96]. A complete genetic characterization of HCC patients could improve the clinical use of mTOR inhibitors, alone or in combination.

Rapalogs and TOR-KIs are associated with feedback loops that promote the (re)activation of these signaling pathways, thus limiting their therapeutic use. The simultaneous inhibition of both, mTORC1 and MAPK, or mTORC1/mTORC2 and RTKs, has been shown to enhance the effectiveness of each compound separately in preclinical models (in vitro and in vivo) [84,89]. The concomitant inhibition of MEK and AKT in mice co-expressing activated forms of c-MET and AKT resulted in a stronger inhibition of tumor progression whereas neither sorafenib nor regorafenib showed any efficacy [97].

Recently, two-step senescence-focused therapies have been proposed as anti-cancer therapy [98]. In this approach, the first drug induces cell vulnerability (senescence in this case), which is exploited by the second agent. In liver cancer, Wang et al. [99] have shown that the TOR-KI AZD8055 [100] exhibits a greater activity in senescent cells as compared with proliferating cells. In this setting, XL413 acted as a second hit to inhibit the DNA-replication kinase CDC7, which selectively induces senescence in liver cancer cells with mutations in TP53 or its upstream regulators. Senescence induction by XL413 disrupted the feedback reactivation of the mTOR signaling caused by the mTOR inhibitor while increasing the apoptosis susceptibility of cancer cells to AZD8055. These observations were consistent with mouse models of HCC and revealed that a combination of XL413 and AZD8055 would be more effective than sorafenib.

Altogether, these results highlight two critical points to select the best therapeutic strategy in HCC. First, to define the components and relationships of the signaling pathways involved in tumor formation, and second, to know the genomic mutations affecting each patient. This knowledge would allow the tailoring of antiproliferative agents according to each patient and tumor characteristics, thus resulting in more effective and safer therapeutic approaches.

## 5. Targeting mTOR Signaling in HCC: From the Bench to the Clinic

The anti-proliferative properties of mTOR inhibitors in preclinical models would be of high value in patients with HCC as they would allow to prevent tumor recurrence after potentially curative therapies (i.e., percutaneous tumor ablation, surgical resection or liver transplantation), or to delay tumor progression in patients with a more advanced disease. Unfortunately, clinical studies are scarce and with reduced methodological quality. Most of the studies are single-centre retrospective observations and those with a prospective and/or randomized design have failed to prove any benefit in terms of recurrence free survival and/or overall survival.

In patients with unresectable or metastatic HCC, the mTOR inhibitor everolimus has been studied in combination with sorafenib. A phase I randomized trial was designed to explore the maximal dose of everolimus tolerated in combination with the standard dose of sorafenib (i.e., 400 mg bid). A total of 30 patients with compensated liver cirrhosis Child Pugh class A were included. The maximal tolerated dose of everolimus was only 2.5 mg per day. The combination of everolimus and sorafenib did not result in improved overall survival. The authors concluded that a phase II trial was not justified since a biologically active anti-proliferative dose of everolimus cannot be achieved in patients with liver cirrhosis, even in those with a preserved liver function [101]. Despite this, a phase II multicenter trial involving 106 patients with unresectable or metastatic HCC and compensated liver cirrhosis was conducted [102]. Patients were randomly assigned to receive sorafenib alone (400 mg bid) or sorafenib (400 mg bid) in combination with everolimus (5 mg od). Progression-free survival rates at 12 months were identical in patients with and without everolimus (68% vs. 70%), but severe adverse events (grade 3–4) were more frequent in the everolimus arm (86% vs. 72%). The authors confirmed that a clinically meaningful inhibition of the mTOR pathway in patients with liver cirrhosis cannot be achieved by using everolimus because of its toxicity. Finally, the sequential use of everolimus in patients with advanced HCC who experience tumor progression under sorafenib was also unable to prolong survival [103].

In earlier stages of HCC, when the tumor is confined to the liver, liver transplantation may be the best therapeutic option as it allows for curing both, the tumor and the underlying liver cirrhosis [104]. However, tumor recurrence rates after liver transplantation are 10% to 15% when HCC is under the so called Milan criteria (i.e., a single tumor less than 5 cm or up to three nodules, less than 3 cm each, without macrovascular invasion) or even higher if more nodules with increased diameter are allowed [91,105]. Since HCC recurrence after liver transplantation is associated with a dismal prognosis, the use of mTOR inhibitors as adjuvant therapy is particularly attractive. Their dual effect as immunosuppressants and antiproliferative agents would prevent graft rejection and, theoretically, would decrease the risk of HCC recurrence [106]. In liver transplantation, there are two mTOR inhibitors approved to be used as immunosuppressive agents. Sirolimus (or rapamycin) entered the market first but the initial phase II randomized trial in liver transplantation had to be prematurely stopped because the experimental arm consisting of sirolimus plus tacrolimus resulted in an increased risk of graft loss and sepsis [107]. A subsequent network meta-analysis of randomized trials have demonstrated that sirolimus-based immunosuppression is associated with increased mortality rates as compared with other protocols and therefore there are concerns about using sirolimus as a first-line therapy in patients receiving a liver transplantation [108]. Everolimus has an improved safety profile as compared with sirolimus. Several randomized trials have shown that, in combination with calcineurin inhibitors, everolimus allows for a more effective preservation of kidney function in liver transplant recipients [109,110]. Regarding prevention of HCC recurrence, several retrospective single-center studies, further meta-analyzed, have reported reduced tumor recurrence rates in patients receiving mTOR inhibitors-based immunosuppression [111]. However, the only prospective observational study (using everolimus) [112] and the only randomized trial (using sirolimus), which was sufficiently powered (*n* = 525) and with prolonged follow-up, have failed to demonstrate any benefit in reducing tumor recurrence rates after liver transplantation [113]. This gap between retrospective and prospective data may be explained again by safety issues [114]. Those patients with severe adverse events leading to significant dose reduction or drug withdrawal, which may account for up to 25% of patients according to the registry trial of everolimus [109], are systematically excluded from retrospective series, leading to a selection bias. In other words, retrospective studies only account for patients with adequate drug tolerance who also receive increased dosage of mTOR inhibitors whereas the results of prospective/randomized studies are diluted because of intolerant and non-compliant patients [114]. The use of everolimus-based immunosuppression would be justified in liver transplant patients with HCC showing poor prognostic histological features such as microvascular invasion in whom the risk of adverse events would be overwhelmed by a theoretical advantage of reduced tumor recurrence rates [64].

In light of the available evidence, mTOR inhibitors (i.e., sirolimus and everolimus) have not been approved to treat or to prevent HCC and they do not form part of the routine clinical armamentarium. In clinical practice, everolimus is used as an immunosuppressive agent in liver transplant patients for a more effective preservation of kidney function. Many institutions have implemented everolimus-based immunosuppression in patients with HCC undergoing liver transplantation in order to prevent tumor recurrence, even in the absence of solid scientific supporting evidence. There is a need for more potent mTOR inhibitors and with an improved safety profile. The role of second-generation TOR-KIs in HCC is still to be explored. Since HCC occurs mainly in the context of chronic liver disease, randomized controlled trials may be conducted in patients with liver cirrhosis to study efficacy and safety.

## 6. Conclusions

The mTOR signaling pathway is involved in cell metabolism and proliferation, which are key hallmarks of cancer and allow for HCC development, progression and spreading. The mTOR pathway inhibition exerts a pronounced antiproliferative effect in vitro and in animal models of HCC. However, clinical studies, especially those prospective and/or randomized, have failed to demonstrate any capacity of mTOR inhibitors to hamper tumor progression or to prevent tumor recurrence after potentially curative therapies such as liver transplantation. The gap between experimental and clinical observations may be partially explained by the dose-dependent toxicity of mTOR inhibitors in patients with chronic liver disease, which frequently motivates a significant dose reduction or drug withdrawal. There is a need to develop more potent mTOR inhibitors and with an improved safety profile in order to obtain a true antiproliferative (and clinically meaningful) effect in patients with HCC. A detailed molecular profiling of HCC will probably become a powerful tool to select patients with HCC to receive mTOR inhibitors as part of a true personalized medicine.

## Figures and Tables

**Figure 1 ijms-21-01266-f001:**
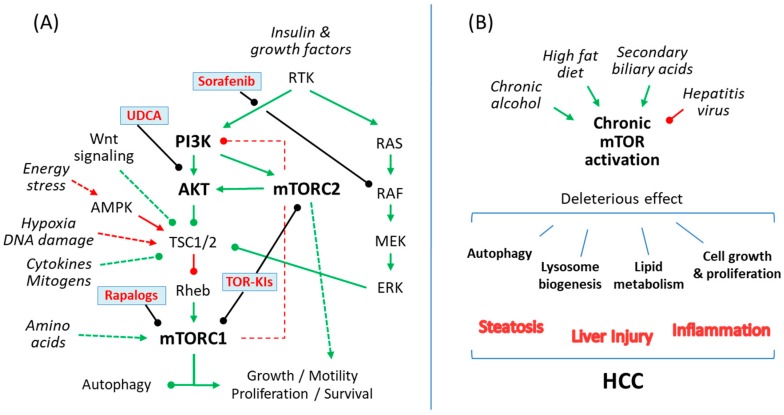
(**A**) A simple scheme of the mammalian target of rapamycin (mTOR) signaling pathway showing action sites for rapalogs and currently available compounds for patients with chronic liver disease, like sorafenib or ursodeoxycholic acid (UDCA), which have been shown to prevent liver damage by interfering with the mTOR pathway. (**B**) Common causes of cirrhosis and HCC such as alcoholic liver disease, chronic viral hepatitis (**B**,**C**), non-alcoholic steatohepatitis and cholestatic liver diseases exert liver damage by causing an abnormal regulation of the mTOR signaling pathway in hepatocytes, which in turn regulates cellular lipid metabolism, autophagy and lysosome biogenesis or cellular growth and proliferation. Green, indicates mTOR pathway activation; Red, indicates mTOR pathway repression; arrowhead, activation; roundhead, inhibition.

**Table 1 ijms-21-01266-t001:** Chemical compounds affecting the mTOR signaling pathway currently used or under investigation in chronic liver diseases and hepatocellular carcinoma (HCC).

Drug Name	Pharmacological Group	Primary Target(s)	Licensed for	Pros and Cons
Sorafenib	Multikinase inhibitor	VEGFR, PDFGR, c-Kit, c-RAF and B-RAF	Standard of care for advanced unresectable HCC	Reduced tolerability, limited efficacy. Highly toxic when combined with mTOR inhibitors.
Rapamycin (sirolimus) and everolimus	mTOR inhibitor	Intracellular receptor FKBP12. Inhibition of mTORC1 (and mTORC2 to a lesser extent)	Immunosuppression after liver transplantation in combination with calcineurin inhibitors	Inefficacious in monotherapy for advanced HCC. Potential effect as adjuvant therapy after liver transplantation to prevent tumor recurrence in selected patients.
mTOR kinase inhibitors	mTOR inhibitor (ATP-competitive)	mTOR kinase domain. Inhibit both mTORC1 and mTORC2	Under investigation for advanced HCC	Ongoing clinical trials in HCC. Safety and efficacy still to be determined.
UDCA (and derivatives)	Bile acid	mTOR signalling pathway	Cholestatic diseases	Anti-tumor effect in preclinical models of HCC. Very well tolerated. No proven benefit for HCC in humans.

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
