# Peer review of "Activation of mTOR Signaling Pathway in Hepatocellular Carcinoma"

_ijms, 2020, doi:10.3390/ijms21041266_

Round 1
Reviewer 1 Report
Review of review paper titled: “Activation of mTOR signaling pathway in hepatocellular carcinoma”.
The purpose of the review was to “analyze the most recent scientific evidence while providing some insights to understand the gap between experimental and clinical studies focusing on the role of the mTOR pathway in the pathogenesis and biological aggressiveness of HCC”. In my opinion the purpose of the work has been achieved, and knowledge conveyed in a comprehensible way. However, the separate chapters 2. and 3. describing mTOR signaling pathway in general overview and regulation of the mTOR system in liver diseases and HCC should be supplemented with table(s) and at least one diagram summarizing new discoveries in the subject. It is not clear for me what is really new in the topic.The review is very interesting, also from clinical point of view, and may carry practical importance. This review cites all relevant and the latest literature in the field. The conclusions and future perspectives drawn are comprehensible. Therefore I support publication. I have no substantive (meritoric) critical comments. However, I have little comments about the text of the work. It would be better for the reader to have the explanation of abbreviations the first time the authors use it, e.g. line 48-49, VEGF/VEGF?, line 77 - AMPK, AKT, line 162 –IKKb/TSC1, etc. Please check also some editorial or stylistic errors: line 74 – I don’t understand the term “their membrane receptors and tyrosine kinases (RTKs)?” Receptor Tyrosine Kinases are from the family of cell-surface (membrane) receptors; line 85 –“in cancer”; line 116- alcohol binging; line 122 citations should be corrected [34,35];
Author Response
Thank you for your constructive comments.
To summarize new discoveries in a table is hard given the wide extent of the reviewed topic. In turn, we have decided to include a new figure, which summarizes the key components of the mTOR pathway and their interaction with the MAPK/ERK pathway, together with regulation points and targets of rapalogs discussed subsequently in the text. We believe that this figure captures the content of chapters 2 and 3 in a graphical way and will ease the reading of subsequent sections. We have also added a table including the properties of available drugs with mTOR inhibition capacity.
In addition, we have included the explanation of abbreviations whenever needed as pointed out by the reviewer. We have also edited the manuscript to correct typos as suggested.
Reviewer 2 Report
Excellent review of literature!
Author Response
Thank you for your positive evaluation. We have edited the manuscript for minor corrections and language polishing.
Reviewer 3 Report
The manuscript systematically described the carcinogenic mechanisms based on the activation of mTOR signaling pathway in hepatocellular carcinoma (HCC) and also therapeutic strategies based on targeting mTOR signaling in HCC treatment. The information given in this review paper is in detailed and comprehensive. The manuscript can be published after 2 minor revisions to improve the readability:
A Figure illustrating the mTOR signaling pathway and its regulation in liver disease and HCC as described from line 58-line 192 should be provided by authors. The Figure should instantly make reader understand the interaction between mTOR and other molecules such as PI3K/AKT and MARK/ERK etc. and also their involvement in the initiation of liver diseases and HCC. A Table summarizing the therapeutic strategies targeting mTOR in HCC treatment from Line 193 to Line 378 should be provided by authors. The Table should list the following information including: the name of pharmacological compounds with anti-mTOR activity (e.g. either monotherapy or combined therapy); the name of cell lines used in pre-clinical models; the brief action mechanisms, the weakness of the therapeutic agents, and the phage of clinical trials (e.g. early stage of HCC or after liver transplantation).Author Response
Thank you for your comments. As suggested, we have included a new figure, which summarizes the key components of the mTOR pathway and their interaction with the MAPK/ERK pathway, together with regulation points and targets of rapalogs discussed subsequently in the text. We have also included a table to summarize the available drugs with an mTOR inhibition capacity. In the text, we have quoted a recent paper Yu et al. (BBA - Reviews on Cancer 1871 (2019) 379–391), which contains detailed information of ongoing clinical trials with mTOR inhibitors alone or in combination with other drugs in HCC and after liver transplantation.
Reviewer 4 Report
This is a good review summarized the relationship between mTOR signaling pathway and HCC. Only two minor concerns here:
1, A figure describing the mTOR singling pathway and its interaction with normal and pathogenic metabolism should be added to make readers better understand the text.
2, Some abbreviations should be given a full name in the text. Such as ROR in line 49 etc.
Author Response
We have introduced all abbreviations as suggested by the reviewer. We have added a new figure, which summarizes the key components of the mTOR pathway and their interaction with the MAPK/ERK pathway, together with regulation points and targets of rapalogs discussed subsequently in the text. We believe that this figure captures the content of chapters 2 and 3 in a graphical way and will ease the reading of subsequent sections.